# Resolving stem and progenitor cells in the adult mouse incisor through gene co-expression analysis

**Kerstin Seidel[1], Pauline Marangoni[1], Cynthia Tang[1], Bahar Houshmand[1], Wen Du[1], Richard L Maas[2], Steven Murray[3], Michael C Oldham[4,5]\*, Ophir D Klein[1,5,6]\***

[1]Department of Orofacial Sciences and Program in Craniofacial Biology, University of California, San Francisco, San Francisco, United States; [2]Division of Genetics, Department of Medicine, Brigham and Women's Hospital, Harvard Medical School, Boston, United States; [3]The Jackson Laboratory, Bar Harbor, United States; [4]Department of Neurological Surgery, University of California, San Francisco, San Francisco, United States; [5]The Eli and Edythe Broad Center of Regeneration Medicine and Stem Cell Research, University of California, San Francisco, San Francisco, United States; [6]Department of Pediatrics and Institute for Human Genetics, University of California, San Francisco, San Francisco, United States

**\*For correspondence:**
michael.oldham@ucsf.edu (MCO);
ophir.klein@ucsf.edu (ODK)

**Competing interests:** The authors declare that no competing interests exist.

**Abstract** Investigations into stem cell-fueled renewal of an organ benefit from an inventory of cell type-specific markers and a deep understanding of the cellular diversity within stem cell niches. Using the adult mouse incisor as a model for a continuously renewing organ, we performed an unbiased analysis of gene co-expression relationships to identify modules of co-expressed genes that represent differentiated cells, transit-amplifying cells, and residents of stem cell niches. Through in vivo lineage tracing, we demonstrated the power of this approach by showing that co-expression module members *Lrig1* and *Igfbp5* define populations of incisor epithelial and mesenchymal stem cells. We further discovered that two adjacent mesenchymal tissues, the periodontium and dental pulp, are maintained by distinct pools of stem cells. These findings reveal novel mechanisms of incisor renewal and illustrate how gene co-expression analysis of intact biological systems can provide insights into the transcriptional basis of cellular identity.
DOI: https://doi.org/10.7554/eLife.24712.001

## Introduction

To maintain homeostasis, adult tissues must replace cells that have completed their life cycle. An emerging model for studying adult mammalian tissue renewal is the rodent incisor, which grows continuously throughout the animal's life. As with many renewing organs, the differentiated cell types of the rodent incisor have a limited life span and are lost over time. A number of cell types, including the ameloblasts and odontoblasts that secrete the mineralized enamel and dentin, respectively, are constantly generated by progenitors located at the proximal end of the tooth (*Figure 1A,B*). These cells replenish the tissues that are lost from the distal end of the tooth due to abrasion during gnawing. In the epithelial compartment, stem cell progeny leave the niche, known as the labial cervical loop (laCL), as they begin the process of differentiation, and they then enter a transit-amplifying (T-A) zone and proliferate (*Kuang-Hsien Hu et al., 2014*). These cells then differentiate, secrete matrix, and finally undergo apoptosis, all the while gradually advancing towards the distal tip of the organ. This linearity, akin to a conveyor belt, makes the incisor a useful model system to study adult epithelial tissue homeostasis, as tissue renewal occurs in an easily-observed proximo-distal fashion,

**eLife digest** To maintain healthy tissues and organs in adult animals, the cells that die or become damaged need to be replaced. This process is made possible by adult stem cells, which can divide to produce more stem cells (via a process called self-renewal) or specialize into other types of cells. This means that stem cells can maintain their own population by self-renewal while continually being able to generate specialized cells that replenish tissues and organs.

Mouse incisor teeth are useful models to understand how adult organs are regenerated because, unlike human teeth, the incisor teeth of mice and other rodents grow continuously throughout the life of the animal. The tip of the mouse incisor is eroded as the animal eats, resulting in the loss of cells. A group of adult stem cells at the base of the tooth produce new cells that then move to the tip to replace the lost cells.

Although virtually all cells in the body have the same set of genes, only small subsets are active in each cell type. It is possible to distinguish cells of different types by their patterns of gene activity. However, little is known about the gene expression patterns that distinguish stem cells and specialized cells in mouse incisors.

Using a technique called gene co-expression analysis, Seidel et al. set out to identify all the genes that are active in stem cells and their descendants at the base of the mouse incisor. The experiments reveal the patterns of activity of thousands of genes, providing a clearer picture of the different cell types present and the biological processes at play. Seidel et al. then used other techniques to identify two genes that can be used as markers to identify distinct types of stem cells in the incisor.

The next steps following on from this work will be to understand in more detail how stem cells behave in renewing the incisor. In the future, these findings may help guide the use of stem cells in regenerating human teeth and other organs.

DOI: https://doi.org/10.7554/eLife.24712.002

whereby cells at increasingly advanced stages of maturation are found at progressively more distal locations (*Figure 1A,B*). The organization of the mesenchyme has been less well-studied than the epithelium, but this tissue also has distinct compartments comprised of progenitors that give rise to various differentiated cell types, including the dentin-producing odontoblasts (*Feng et al., 2011*; *Kaukua et al., 2014*; *Zhao et al., 2014*). A tissue complex of mesenchymal origin known as the periodontium wraps the incisor growth region and anchors the tooth in the jaw (*Nanci and Bosshardt, 2006*).

Relatively little is known about the molecular identities of progenitor cells in the incisor or about the signals they use to regulate the production of cell types that are required to maintain homeostasis. The high turnover and short lifespan of differentiated cell types in the incisor indicate that there are active pools of progenitor cells (*Smith and Warshawsky, 1976*, *Smith and Warshawsky, 1975*). In vivo lineage tracing assays have identified *Gli1* and *Bmi1*-expressing populations of stem cells in both the incisor epithelium and mesenchyme (*Biehs et al., 2013*; *Seidel et al., 2010*). Both *Gli1* and *Bmi1* also mark label-retaining cells (LRCs) that divide infrequently and therefore retain BrdU or genetic labels. Within the incisor, LRCs are restricted to the proximal incisor mesenchyme and the proximal part of the laCL and lingual cervical loop (liCL). Additional lineage-tracing studies identified *Sox2* as a stem cell marker in the incisor epithelium but not the mesenchyme (*Juuri et al., 2012*). The properties displayed by *Gli1-*, *Bmi1-* and *Sox2*-expressing cells – slow division kinetics, residence in a discrete niche, and contribution to the differentiation of various lineages – are classically considered to be typical of stem cell populations. To date, only these three markers have been found to label incisor stem cells, and thus a major limitation of the incisor model has been a paucity of markers that clearly distinguish its cell types, including progenitor cells. Indeed, this limitation is not unique to the incisor, as the precise cellular composition of most mammalian organs is still unclear. The ability to clearly distinguish cell types and distinct stages of maturation is an essential prerequisite to understanding renewal and regeneration.

Gene co-expression analysis is a powerful approach for elucidating transcriptional signatures of distinct cell types in heterogeneous tissue samples (*Oldham et al., 2008*). This approach is based on two straightforward ideas: (i) different cell types express different genes, and (ii) the relative

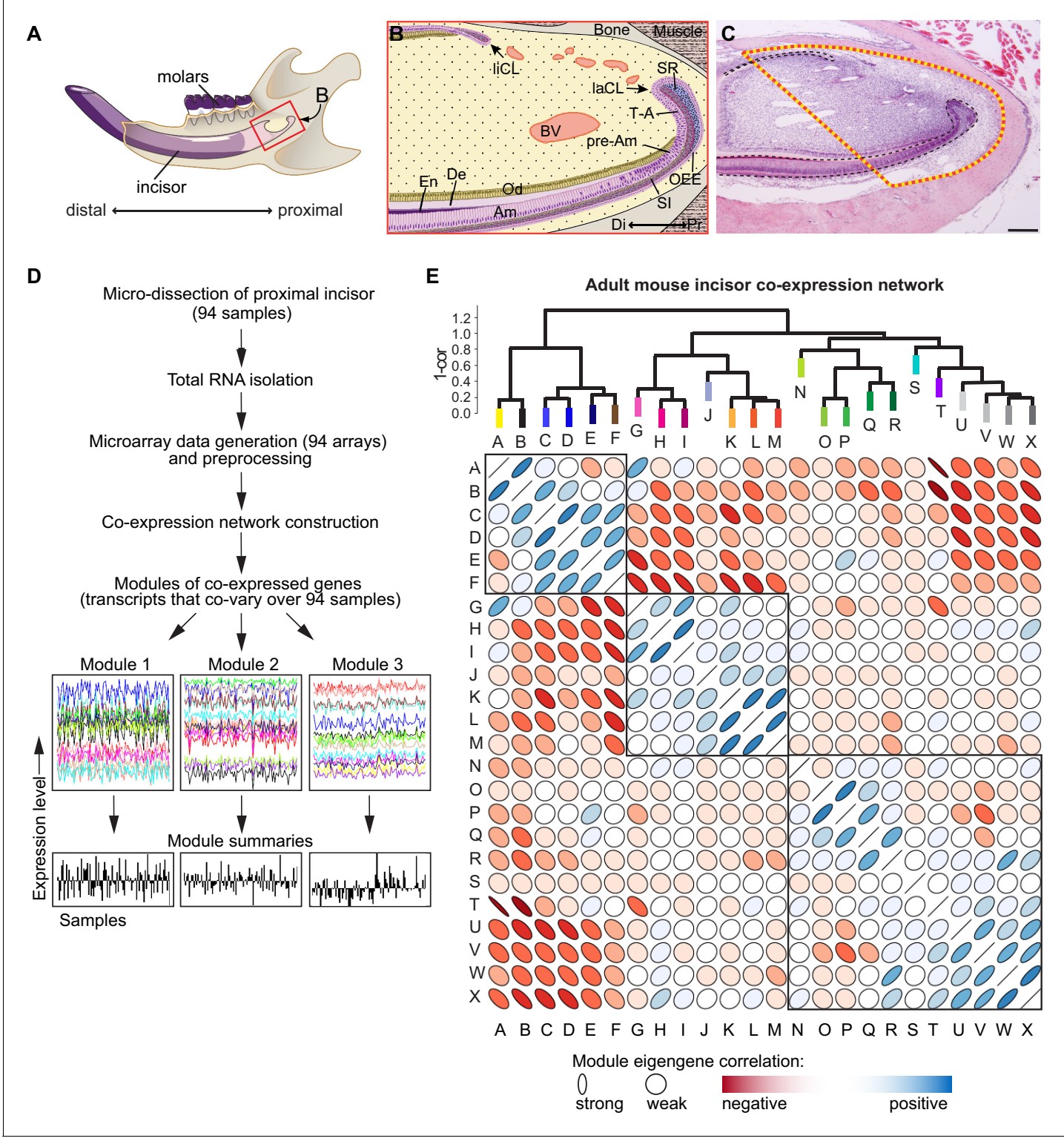

**Figure 1.** Analysis of transcriptional co-variation in adult mouse incisor reveals gene co-expression modules. (**A**) Diagram depicting sagittal view of right lower jaw of a mouse. (**B**) Schematic of the stem cell-containing region in the proximal (Pr) mouse incisor. Stem cell pools that give rise to epithelial cell types (purple) are located in the proximal portion of the lingual and labial cervical loops (liCL, laCL). Mesenchymal cell types of the incisor (yellow), such as the dentin (De)-secreting odontoblasts (Od), are constantly replenished by progenitors located in the mesenchyme between the CLs. Blood vessels (BV) are highly abundant in this region. In the laCL, stem cells located in the outer enamel epithelium (OEE), adjacent proximal stellate reticulum (SR), and stratum intermedium (SI), give rise to highly proliferative transit-amplifying (T–A) cells which, after undergoing mitosis, differentiate along several

*Figure 1 continued on next page*

*Figure 1 continued*

epithelial lineages as the progeny advance to the distal (Di) tip of the incisor and differentiate into enamel (En)-secreting ameloblasts (Am). (C) Hematoxylin and eosin stained sagittal section of the mouse incisor. Dashed line indicates tissue region for which analysis was performed. Scale bar: 200 μm. (D) Workflow for incisor gene co-expression network construction. RNA samples were used to generate genome-wide microarray expression profiles for 94 intact incisor specimens, which were used as input for unsupervised gene co-expression analysis. Gene co-expression modules consist of transcripts that co-vary and therefore have highly similar expression signatures across all samples (x-axis). Examples of three co-expression modules are shown. The characteristic expression pattern of each module is summarized by its first principal component, or module eigengene (module summaries; (*Horvath and Dong, 2008*)). (E) Structure of the incisor co-expression network. Twenty-four gene co-expression modules were identified and hierarchically clustered based on eigengene dissimilarity (1 – cor) using average linkage. The correlation matrix of the module eigengenes is depicted below. Blue and red denote positive and negative correlations, respectively, with stronger correlations denoted by thinner ellipses (*Murdoch and Chow, 1996*). Three main clusters of positively correlated modules are evident from the dendrogram and correlation plot.

DOI: https://doi.org/10.7554/eLife.24712.003

abundance of a given cell type will vary among heterogeneous tissue samples. Therefore, transcript levels of genes that are specifically and consistently expressed by a cell type will appear highly correlated when measured over a large number of biological replicates. We set out to apply this strategy to the proximal adult mouse incisor region, with the goal of identifying transcriptional signatures of progenitor cells and their descendants. We identified modules of co-expressed genes representing differentiated cells, transit-amplifying cells, and residents of stem cell niches. We further demonstrated the power of this approach by using in vivo lineage tracing to define populations of incisor stem cells, and we discovered that two adjacent mesenchymal tissues, the periodontium and dental pulp, are maintained by distinct pools of stem cells. More generally, our data indicate that this strategy provides a useful analytical framework for deconstructing biological systems by identifying recurrent patterns of transcriptional variation driven by large numbers of cells.

## Results

### Transcriptome analysis of the mouse incisor reveals modules of co-expressed genes

We micro-dissected proximal incisor samples from 94 six-week-old female CD1 wild-type mice; each tissue sample was heterogeneous and consisted of the entire range of cell types present in the proximal incisor region (*Figure 1C*). We profiled transcriptomes using Illumina Mouse Ref 8 v2.0 gene expression BeadChip microarrays, which contain 25,697 probes. Following data pre-processing (*Oldham et al., 2012*), we performed genome-wide gene co-expression analysis (*Lui et al., 2014*; *Zhang and Horvath, 2005*) and identified 24 modules of co-expressed genes (termed A-X, *Figure 1D*; *Figure 1E*). The characteristic expression pattern of each module was summarized by its first principal component, or module eigengene (ME) (*Horvath and Dong, 2008*; *Oldham et al., 2008*), and verified by selecting several markers among the highest ranked genes and conducting in situ hybridization (ISH) analysis. Hierarchical clustering of modules based on eigengene dissimilarity revealed distinct subgroups of modules within the dendrogram (*Figure 1E*), suggesting broad themes of transcriptional co-variation in the incisor.

### A subset of gene co-expression modules corresponds to distinct cell lineages

We quantified the similarity between the expression patterns of individual genes and the eigengenes of co-expression modules by calculating the Weighted Gene Co-expression Network Analysis (WGCNA) (*Zhang and Horvath, 2005*) measure of intramodular connectivity, $k_{ME}$, for all genes with respect to all modules (*Supplementary file 1*). $k_{ME}$ is defined as the Pearson correlation between the expression pattern of a gene and a ME (*Horvath and Dong, 2008*). Intuitively, the ME summarizes the characteristic expression pattern of genes comprising a module, and $k_{ME}$ quantifies the extent to which individual genes track this pattern. $k_{ME}$ can therefore be used to identify individual genes that best represent a module and mark particular cell types or biological processes (*Oldham et al., 2008*). To determine how these modules mapped to cell types of the incisor, we initially examined the top 15 genes ranked by $k_{ME}$ and conducted an extensive ontology search using

literature available in PubMed. We also examined genes reported to be expressed during late development and postnatal life in the 'Gene Expression in Tooth' database (http://bite-it.helsinki.fi/).

This analysis immediately suggested that Module A (*Figure 1E*; *Figure 2A*) represents a transcriptional signature of ameloblasts, which are enamel-producing differentiated cells derived from epithelial stem cells in the laCL (*Figure 1B*). For example, both *Lamb3* ($k_{ME.A}$rank = 4) and *Lamc2* ($k_{ME.A}$rank = 10) encode subunits of Laminin 5 that are expressed by ameloblasts in developing mouse incisors (*Yoshiba et al., 2000*). Expression of *Enam* ($k_{ME.A}$ rank = 11) is also restricted to ameloblasts (*Kuang-Hsien Hu et al., 2014*). For *Lamc2* and *Enam*, multiple microarray probes targeting these transcripts had $k_{ME}$ ranks for Module A within the top 0.5% of all probes on the microarray (*Supplementary file 1*). Similarly, *Ambn* and *Amelx*, which encode enamel matrix proteins that are widely used as ameloblast markers (*Lee et al., 1996*; *Snead et al., 1988*), were also strongly associated with Module A ($k_{ME.A}$ ranks of 107 and 118, respectively; *Supplementary file 1*). Another gene in this module was *Dact2* ($k_{ME.A}$rank = 192), which encodes a transcription factor-binding protein whose expression has previously been shown to be restricted to the dental epithelium during molar development (*Kettunen et al., 2010*).

Further investigation showed that Module C represents a transcriptional signature of odontoblasts, which are the dentin-secreting cells comprising the outer layer of the dental pulp. For example, *Bglap1* and *Bglap2* were among the top-ranked genes for Module C (*Figure 2B*), and these are expressed by mesenchymally derived pre-odontoblasts and odontoblasts (*Bronckers et al., 1987*). *Phex* ($k_{ME.C}$ rank = 1) is expressed in developing odontoblasts (*Ruchon et al., 1998*), and *Dspp*, a known marker of odontoblasts (*Bègue-Kirn et al., 1998*; *D'Souza et al., 1997*), was strongly associated with Module C ($k_{ME.C}$ rank = 190). Similarly, *Kazald1*, which is expressed by secretory odontoblasts (*James et al., 2004*), was also associated with Module C ($k_{ME.C}$ rank = 266).

We next asked if module organization could predict novel markers of distinct cell types, beginning with differentiated cell types such as ameloblasts and odontoblasts. We used immunohistochemistry and ISH to examine expression of genes with high $k_{ME}$ for those modules that, to our knowledge, have not been previously implicated in ameloblast or odontoblast biology. As shown in *Figure 2C*, *Tmem108* ($k_{ME.A}$ rank = 1), SOX21 ($k_{ME.A}$ ranks = 5, 15, 1302), and *StarD10* ($k_{ME.A}$ rank = 8) all showed robust and specific expression in the ameloblast lineage. These data demonstrate that Module A consists of genes that are predominantly expressed in the ameloblast lineage of the adult mouse incisor.

Next, we investigated expression patterns of genes that were strongly associated with Module C, to determine if the module organization was an effective tool to predict novel markers of odontoblast identity (*Figure 2D*). As expected, expression of *Phex* ($k_{ME.C}$ rank = 1) was restricted to odontoblasts (*Ruchon et al., 1998*). Expression of *Bglap1*, *Blgap2*, and *Bglap-rs1* ($k_{ME.C}$ ranks = 6, 9, 10, 14) was detected with a probe for an mRNA sequence shared by all three genes, confirming specificity to the odontoblast lineage. Expression of *Sag* ($k_{ME.C}$ rank = 7) was also restricted to odontoblasts, while expression of *Lox* ($k_{ME.C}$ rank = 8) showed a more complex pattern, with robust expression in odontoblasts but additional expression in regions where T-A cells are located in epithelium and mesenchyme (arrow and asterisk in *Figure 2D*). This discrepancy may reflect biological or technical sources of variability such as additional *Lox* isoforms and/or non-specific targeting by microarray or ISH probes. Overall, these results indicate that Module C consists of genes that are predominantly expressed in the odontoblast lineage.

To compare the distributions of predicted expression specificity for ameloblasts (*Z*.ameloblast) and odontoblasts (*Z*.odontoblast), we generated standardized, genome-wide histograms of $k_{ME}$ values for each module (*Figure 2E*) (*Lui et al., 2014*). We observed that known markers of ameloblasts possessed *Z*.ameloblast >> *Z*.odontoblast, and vice versa. These results indicate that the expression signatures captured by Modules A and C are both sensitive and specific: canonical markers of each cell type have high $k_{ME}$ values for the appropriate module and lower $k_{ME}$ values for the inappropriate module. Importantly, the vast majority of genes with the highest $k_{ME}$ values for these two modules, which are likely to play central roles in the establishment and maintenance of the functional identities of these cell types (*Lui et al., 2014*; *Oldham et al., 2008*) have not been characterized in the ameloblast or odontoblast lineages.

Taken together, these findings establish that gene co-expression analysis of heterogeneous tissue samples can discern and predict transcriptional signatures of the two principal secretory cell types of the adult mouse incisor. We also identified gene co-expression modules corresponding to other

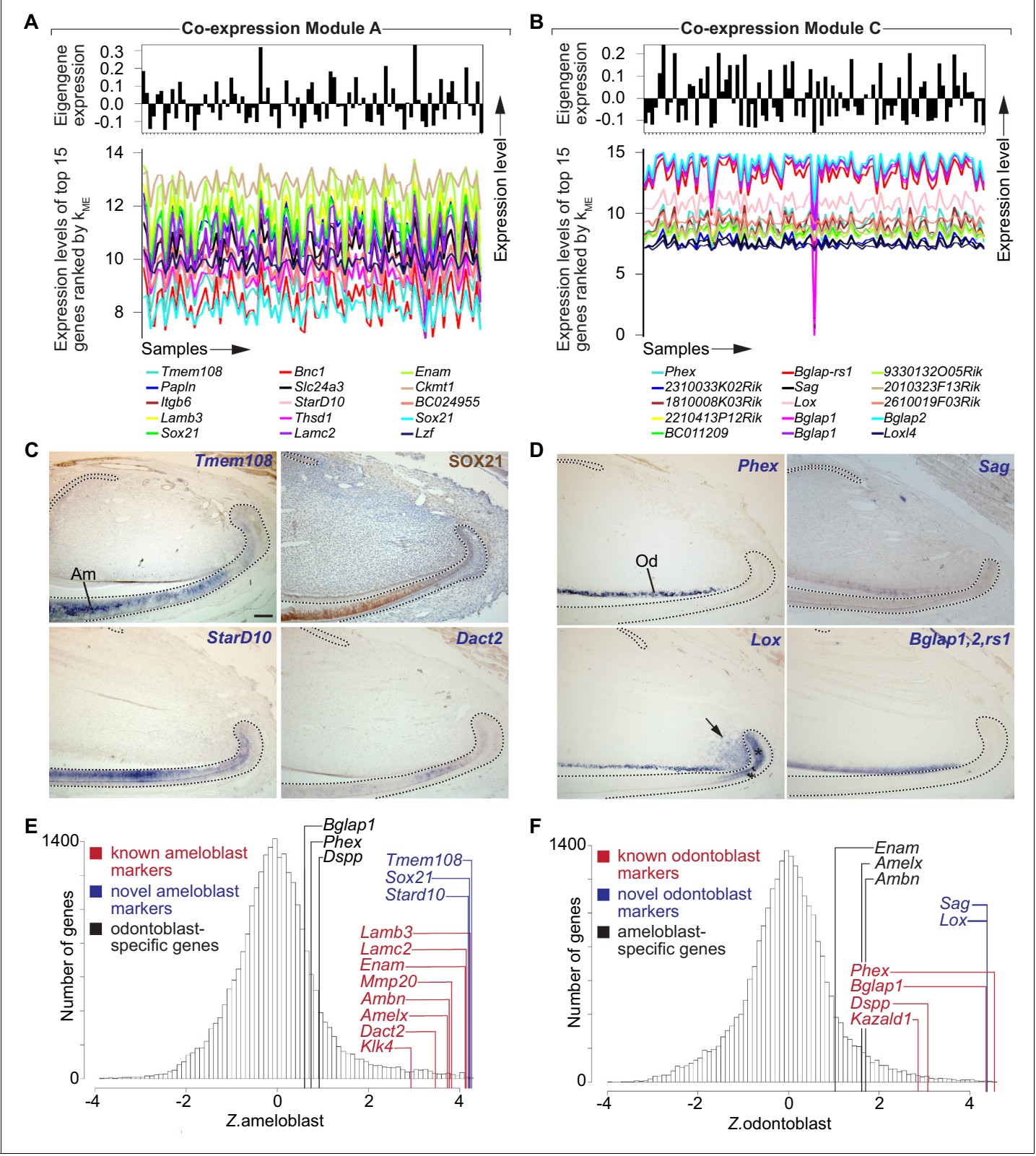

**Figure 2.** Identification of gene co-expression modules corresponding to distinct lineages of differentiated stem cell progeny. (A,B) Snapshots of gene co-expression Modules A and C. Top: the module eigengene (first principal component) summarizes the characteristic expression pattern of genes in each module. Bottom: expression patterns of the top 15 genes ranked by $k_{ME}$ values for each module. (C) Immunohistochemistry (top-right panel) and in situ hybridization (remaining panels) confirms ameloblast-specific expression for genes in Module A. (D) In situ hybridization confirms expression in

*Figure 2 continued on next page*

*Figure 2 continued*

odontoblasts for genes in Module C. Arrow and asterisk denote additional expression domain of *Lox* in mesenchymal and epithelial T-A region, respectively. (E,F) Genome-wide distribution of predicted ameloblast (E) and odontoblast (F) expression specificity. Dashed lines in (C,D) delimit ectodermal epithelium. Am, ameloblasts; Od, odontoblasts. Scale bar: 100 µm.

DOI: https://doi.org/10.7554/eLife.24712.004

The following figure supplement is available for figure 2:

**Figure supplement 1.** Identification of additional modules corresponding to differentiated and differentiating cell types.

DOI: https://doi.org/10.7554/eLife.24712.005

differentiated cell types that are present in this biological system (*Figure 2—figure supplement 1*). For example, over-representation analysis with cell type-specific gene sets revealed that Module J was significantly enriched with experimentally validated markers of oligodendrocytes (p<10$^{-8}$, *Figure 2—figure supplement 1B–E*), suggesting that this module corresponds to a transcriptional signature of Schwann cells. Similarly, Module S was significantly enriched with experimentally validated markers of skeletal muscle cells (p<10$^{-12}$, *Figure 2—figure supplement 1B*). It appears that this highly specific module resulted from contamination of a small number of tissue samples by cells from the muscle tissue surrounding the jawbones. These modules corroborate the ability of gene co-expression analysis to isolate distinct transcriptional signatures of differentiated cell types from heterogeneous tissue samples in silico while simultaneously providing a broad perspective on the cellular composition of biological systems.

## A subset of gene co-expression modules corresponds to cellular differentiation states

Although some gene co-expression modules clearly corresponded to specific cell lineages, others were less obvious. To determine which cells in the incisor were responsible for producing the transcriptional patterns captured by these modules, we analyzed expression patterns for at least three of the top 15-ranked members of each module. As a starting point, we chose modules that were strongly positively correlated with Modules A and C and therefore clustered in the left branch of the incisor network dendrogram (*Figure 1E*). Interestingly, expression of *Cdkn1a* ($k_{ME.B}$ ranks = 2, 3, 7), *Smox* ($k_{ME.B}$ranks = 4, 5, 9, 31, 68), and *Atp2b* ($k_{ME.B}$ rank = 8), which were among the highest-ranked members of Module B (*Supplementary file 1*), was detected in epithelium-forming cell types of the ameloblast, stratum intermedium, and odontoblast lineages located distal to the laCL (*Figure 2—figure supplement 1G–I*). Given that a number of genes are expressed in all three lineages during the differentiation and secretory stages of tooth development (http://bite-it.helsinki.fi/), this finding was not surprising. The expression patterns of these genes, which included factors involved in cell cycle exit, are consistent with the location of Module B in the dendrogram and suggest a close relationship between genes in this module with those that are expressed in the ameloblast and odontoblast lineages. Expression patterns of genes in Modules D and E were similar to those of genes contributing to Module C (data not shown), and genes contributing to Module F, including *Tgfbi* ($k_{ME.F}$ ranks = 1, 2), *Fgfr3* ($k_{ME.F}$ ranks = 11, 99), and *Nes1* ($k_{ME.F}$ ranks = 28, 49), were expressed in the distal dental pulp mesenchyme, but not in the proximal-most pulp cells (*Figure 2—figure supplement 1J–L*). Thus, expression of genes contributing to Modules A-F was predominantly detected in regions distal to the stem cell niches in the epithelial cervical loops and in the mesenchyme located between these epithelial regions. The clustering of Modules A-F may reflect correlated cellular abundance among these regions, as dissections that include a greater representation of the epithelial niches are more likely to include a greater representation of the mesenchyme between them.

## Transit-amplifying cells are represented by multiple modules

Functional enrichment analysis indicated that Module U, the second largest co-expression module, was enriched for genes expressed during the mitotic phase of the cell cycle (p=3.1×10$^{-24}$). We therefore analyzed expression patterns for genes in this module as well as other modules with which it was positively correlated (*Figure 1E*, *Figure 3A*). As predicted by the functional enrichment analysis, expression of *Pbk* ($k_{ME.U}$ rank = 1), *Ncaph* ($k_{ME.U}$ rank = 2), and *Cdca2* ($k_{ME.U}$ rank = 3), the three

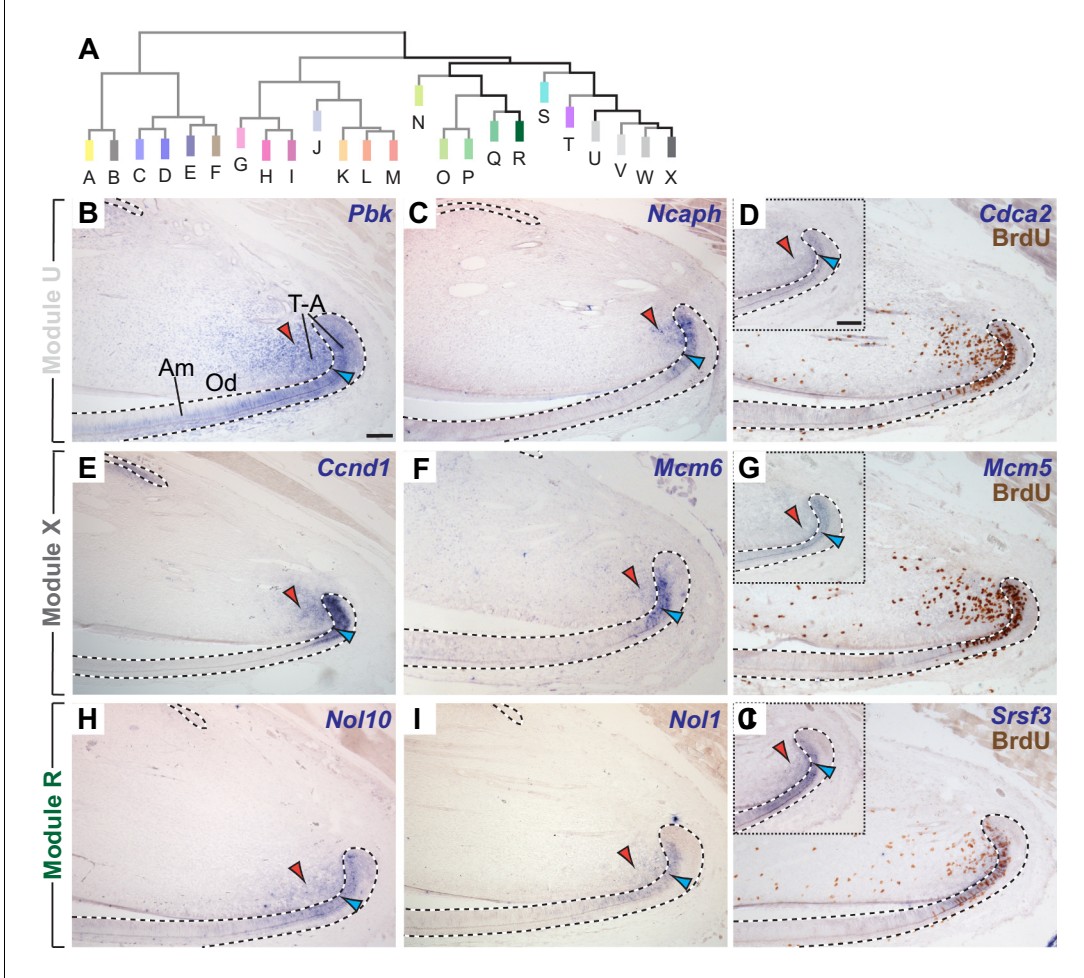

**Figure 3.** The incisor gene co-expression network contains several modules of genes co-expressed in transit-amplifying (T–A) cells. (A) Highly ranked genes contributing to modules clustered in the right-hand portion of the dendrogram are expressed in regions with actively proliferating epithelial and mesenchymal T-A cells. (B–D) mRNA expression of genes highly ranked in module U is detected in T-A cells in the incisor epithelium (blue arrowhead) and mesenchyme (red arrowhead). (E–G) In situ hybridization for highly ranked genes contributing to Module X. (H–J) Transcription of genes in Module R is restricted to T-A cells. In situ hybridization and antibody staining against BrdU (D,G,J) confirm expression of *Cdca2*, *Mcm5* and *Srsf3* in proliferating cells. Insets show mRNA expression prior to detection of BrdU on same tissue section. Dashed lines delimit ectodermal epithelium. Am, ameloblasts; Od, odontoblasts, T-A, transit-amplifying cells. Scale bars: 100 µm, C-J as in B; insets in G, J as inset in D.

DOI: https://doi.org/10.7554/eLife.24712.006

The following figure supplements are available for figure 3:

**Figure supplement 1.** Identification of co-expression modules enriched for transit-amplifying (T–A) cell-specific genes.
DOI: https://doi.org/10.7554/eLife.24712.007
**Figure supplement 2.** Functional annotation of enriched Gene Ontology (GO) terms for Module clusters N-R, T-X and Module S.
DOI: https://doi.org/10.7554/eLife.24712.008

highest ranked genes contributing to Module U (*Supplementary file 1*), was found in actively proliferating transit-amplifying (T-A) cells in the proximal incisor (*Figure 3B–D*), which can be visualized as BrdU-incorporating cells using immunohistochemistry (*Young et al., 1992*). All three genes were expressed in BrdU-positive, T-A cell-containing regions in both the incisor epithelium and mesenchyme.

Interestingly, T-A cell-specific expression was not restricted to members of Module U. We also identified a T-A cell signature in co-expression Modules V, W, and X, which were the modules most strongly positively correlated with Module U (*Figure 3E–G*, and *Figure 3—figure supplement 1I–L*). Similar to the genes contributing to Module U, expression of genes in Modules V, W, and X labeled proliferating cells in both the epithelium or the mesenchyme in all cases. When we extended our

follow-up analyses to modules that clustered within the same branch of the dendrogram, but that were less strongly correlated to Module U, we discovered that Modules N-R also represented T-A cell-specific expression signatures (*Figure 3H–J*, and *Figure 3—figure supplement 1A–D*). The surprising finding that T-A cells were represented by multiple closely related modules of co-expressed genes reflects the importance of these actively dividing cells as central executors of incisor renewal as well as the ability of the methodology to unravel detailed gene expression profiles within both epithelial and mesenchymal T-A cell regions. To better understand what the T-A cell-specific modules represent, we analyzed enrichment patterns in these modules for Gene Ontology annotations related to cell proliferation (*Figure 3—figure supplement 2*). Whereas Modules T-X were enriched in genes involved in processes characteristic of the M-phase of the cell cycle, Modules N-R were enriched in genes involved in biosynthetic and metabolic processes, suggesting roles during interphase, during which cell growth and DNA replication occur. Together, these data indicate that T-A-specific gene co-expression modules may represent distinct biological processes that are integrated to induce or maintain proliferation. They also suggest that these processes may be temporally segregated in subpopulations of T-A cells. The T-A cell-specific modules provide a platform for elucidating the molecular mechanisms that regulate stem cell progeny during this poorly understood stage of maturation.

## Identification of modules enriched with markers expressed by epithelial progenitors

We next asked if the handful of stem cell markers that have been identified in the adult mouse incisor through in vivo lineage tracing were associated with particular co-expression modules. *Gli1* and *Bmi1* mark stem cell pools in both the incisor epithelium and mesenchyme, whereas *Sox2* exclusively marks epithelial stem cells (*Biehs et al., 2013*; *Juuri et al., 2012*; *Seidel et al., 2010*). Although *Gli1* was not represented by a probe on the microarrays that we used, *Ptch1*, which like *Gli1* reports active Hedgehog signaling and is expressed in the same pattern in the incisor as *Gli1* (*Seidel et al., 2010*), was associated with Module L, and *Sox2* was associated with Module H (*Figure 4A*, *Supplementary file 1*). Expression of *Bmi1* was detected in all specimens but not associated with a co-expression module. *Lgr5*, another gene that recently has been suggested to be expressed by incisor stem cells (*Chang et al., 2013*; *Suomalainen and Thesleff, 2010*), was not detected by the microarray in any of the samples. These results may reflect the limited sensitivity of microarrays for detecting low-expressed transcripts in rare cell populations in heterogeneous tissue samples.

Module H, the co-expression module with which *Sox2* was most strongly associated, also contained *Tbx1* ($k_{ME.H}$rank = 14), which was previously shown to be expressed in the epithelium of developing incisors (*Catón et al., 2009*). We found that *Tbx1* expression is also restricted to the epithelium in the adult incisor, and we detected transcripts in the T-A and pre-ameloblast region as well as in several cells in the proximal region of the laCL (*Figure 4B*) where epithelial stem cells reside. Other genes strongly associated with this putative epithelial progenitor module included *Epha1* and *Prom2* (*Figure 4C*, and *Figure 4—figure supplement 1A*); like *Sox2* (*Juuri et al., 2012*) and *Tbx1*, expression of these genes was restricted to the incisor epithelium and included the proximal regions of the laCL, where we previously detected LRCs. Expression of *Epha1* was found predominantly in stellate reticulum (SR) cells in the laCL and in more distal cells subtending the SR cells adjacent to this region (*Figure 4—figure supplement 1A*). In contrast, expression of *Prom2* appeared to overlap that of *Tbx1* in the T-A cell and pre-ameloblast region as well as in the outer enamel epithelium (OEE) of the laCL (*Figure 4B*). In addition to being a highly-ranked gene for Module H, *Prom2* also showed a strong association to the neighboring Module I. Further investigation of Modules G and I, which were strongly positively correlated with Module H and each other (*Figure 1C*), revealed that multiple genes, including *Epha1* and *Prom2*, associated strongly with two or all three modules (*Supplementary file 1*). Therefore, we extended our expression analysis to include the two neighboring modules and treated the G-H-I group of modules as a clustered unit.

Of the genes primarily contributing to Module G, we analyzed the expression patterns of *Scnn1b* ($k_{ME.G}$ rank = 3), *p63* ($k_{ME.G}$ rank = 5), and *Nkx2-3* ($k_{ME.G}$ rank = 11), the last of which was previously shown to be required for molar development (*Biben et al., 2002*). All three were expressed predominantly in the T-A and OEE regions of the laCL with no or low levels of expression in the SR (*Figure 4D*, and *Figure 4—figure supplement 1B*). Expression was not limited to the CL on the labial side but rather extended distally into both the OEE and differentiated ameloblasts and was

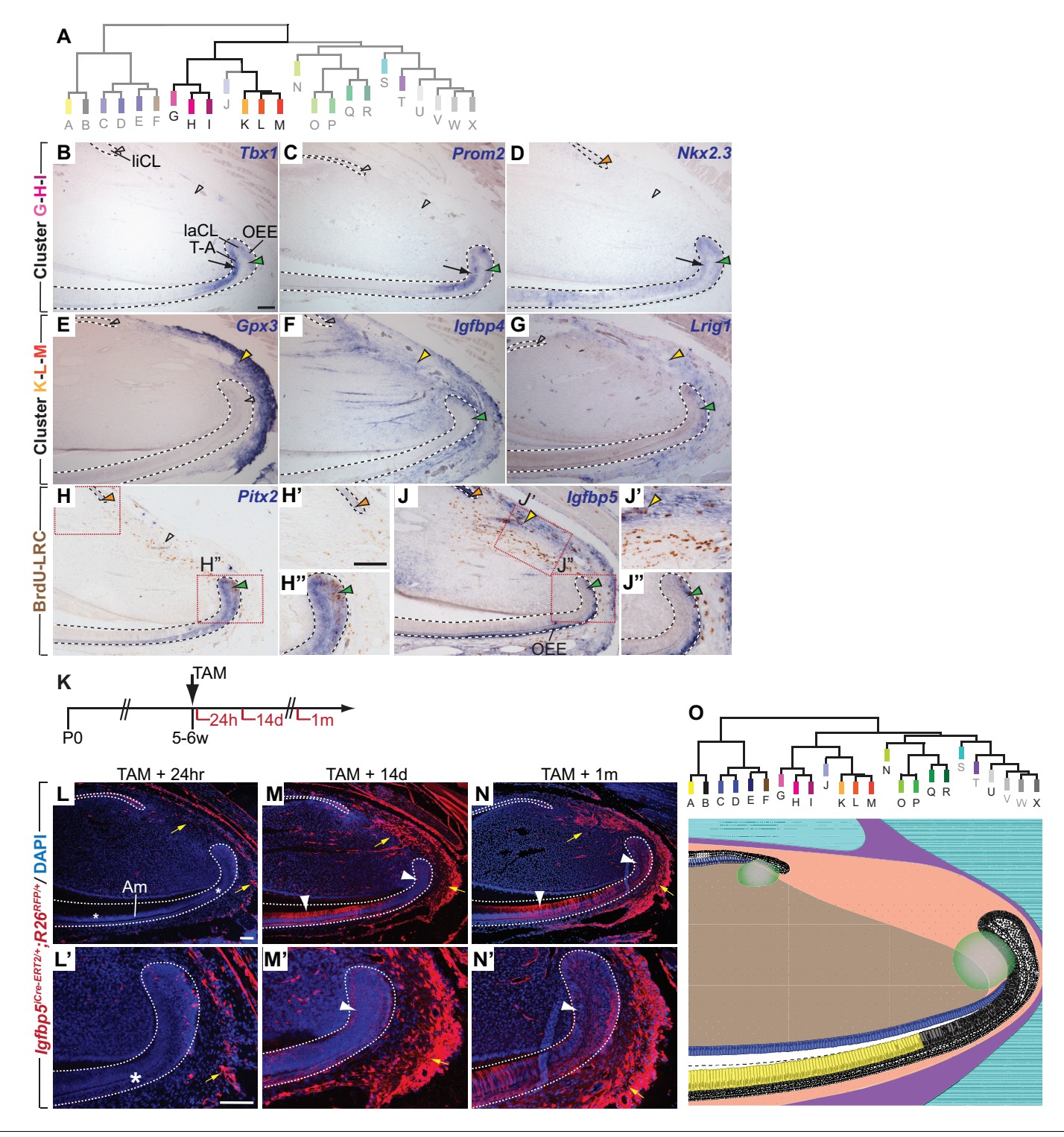

**Figure 4.** Module clusters G-H-I and K-L-M are enriched for candidate epithelial and mesenchymal stem cell markers. (**A**) Dendrogram depicting incisor gene co-expression network with modules featured in panels B-J highlighted. (**B–D**) mRNA expression of genes with high module memberships for modules G, H and I is restricted to the ectodermal epithelium and includes the proximal epithelium of the labial cervical loop (laCL; green arrowheads) and the epithelial T-A region (black arrows). Expression in the lingual cervical loop (liCL) was detected for *Nkx2.3* (orange arrowhead) but not in case of *Tbx1* and *Prom2* (open arrowheads) (**E–G**) Genes contributing to modules K-L-M are predominantly expressed in the proximal incisor mesenchyme (yellow arrowheads). A number of genes affiliated with this cluster show additional transcriptional activity in the proximal laCL (green arrowheads). (**H–H″**) In situ hybridization and antibody staining against BrdU in samples with BrdU-label-retaining cells (LRCs) detects expression of *Pitx2*, which

*Figure 4 continued on next page*

*Figure 4 continued*

contributes to the G-H-I module cluster, in LRCs in the laCL and in the liCL. (**J–J"**) *Igfbp5* is expressed by cells in the liCL, the outer enamel epithelium (OEE), including BrdU-LRCs in the laCL (green arrowhead in J, J"), and a subset of mesenchymal LRCs located close to the periphery of the organ (yellow arrowheads in J and J'). (**K**) Dosing scheme for in vivo lineage analysis of cells expressing *Igfbp5*. (**L–N'**) Lineage tracing of *Igfbp5*-positive cells 24 hr (**L, L'**), 14 days (**M, M'**), and 1 month (**N, N'**) post Tamoxifen treatment. Asterisks indicate the absence of labeled epithelial cells. White arrowheads highlight labeled progeny in the T-A region or amongst differentiated ameloblasts (Am). In the periodontal tissue, labeled cells increase in number over time (yellow arrows). (**O**) Summary of the domains of expression of the modules and module clusters mapped onto a schematic view of the incisor growth region. Color code matches the dendrogram; overlapping modules are represented using corresponding color shading. Module clusters C-D-E, G-H-I, K-L-M, and U-V-W-X are represented by the central hue used for that branch of the dendrogram. Dashed lines delimit ectodermal epithelium. T-A, transit-amplifying cells. Scale bars: 100 μm, C-H and J as in B; H", J', J" as in H'; M, N as in L; M', N' as in L'.

DOI: https://doi.org/10.7554/eLife.24712.009

The following figure supplements are available for figure 4:

**Figure supplement 1.** Identification of co-expression modules enriched for stem cell-specific genes.

DOI: https://doi.org/10.7554/eLife.24712.010

**Figure supplement 2.** Targeting strategy for generation of Tamoxifen-inducible *Igfbp5^{iCreER-T2}* mouse line.

DOI: https://doi.org/10.7554/eLife.24712.011

**Figure supplement 3.** *Acta2*-positive and *Lrig1*-expressing stem cells supply overlapping yet distinct regions in the incisor periodontal compartment.

DOI: https://doi.org/10.7554/eLife.24712.012

also found in the liCL. *Isl1* ($k_{ME.G}$ rank = 33), which was previously shown to be expressed exclusively in the incisor epithelium during tooth development (**Mitsiadis et al., 2003**), was expressed in the same pattern in the laCL as p63, *Nkx2-3*, and *Scnn1b*, with distally extending expression maintained in the OEE and ameloblasts but decreased levels in the pre-ameloblasts (**Figure 3—figure supplement 1C**). The distally extended expression appears characteristic of Module G genes and is reflected in the strong positive correlation of Module G's eigengene signature with that of the ameloblast-specific Module A (**Figure 1E**), whereas the correlations between Modules H and A as well as I and A were weaker (**Figure 1E**).

Expression of *Pitx2* ($k_{ME.I}$rank = 2) in the T-A cell and OEE regions of the laCL widely overlapped with the expression domains of *Prom2* (**Figure 4C**). *Pitx2* expression appeared to be highest in the apical aspect of the laCL, where *Sox2* is expressed at high levels (**Juuri et al., 2012**), and was also present in the SR. In the proximal SR and OEE regions, expression of *Pitx2* was detected in LRCs (**Figure 4H–H"**). However, similar to *Tbx1*, *Epha1* and *Prom2*, expression of *Pitx2* was not detected in the liCL epithelium. *Shh*, which is expressed in epithelial T-A cells, pre-ameloblasts and ameloblasts in the labial incisor epithelium (**Seidel et al., 2010**), was also among the genes contributing to Modules G, H, and I ($k_{ME.G}$rank = 14, $k_{ME.H}$rank = 53, and $k_{ME.I}$rank = 7), which was surprising given the absence of *Shh* expression in the proximal laCL. Thus, expression of all analyzed genes contributing to Modules G, H, and I specifically overlapped with T-A cells in the epithelium and, with the exception of *Shh*, in the proximal laCL.

## Modules K-M are enriched for candidate stem cell markers and markers specific to periodontal tissues

We next focused our attention on modules that appeared to exhibit mesenchymal character. As with Modules G-I, Modules K-M were highly correlated, and several genes showed promiscuity for all three modules (**Figure 1E**, **Supplementary file 1**). Expression analysis of *Gpx3*, which was strongly correlated to all three modules ($k_{ME.K}$rank = 10, $k_{ME.M}$rank = 2, and $k_{ME.M}$rank = 6), *Fbln1* ($k_{ME.L}$ rank = 4), and *Igfbp4* ($k_{ME.L}$ranks = 5,12), showed that all three genes are actively transcribed in the mesenchymal tissue in the proximal part of the incisor (**Figure 4E**, and **Figure 4—figure supplement 1E**). In the case of *Igfbp4*, expression was observed in an additional domain in the proximal laCL epithelium. Similar expression patterns were observed for *Pecam1* ($k_{ME.K}$ rank = 13), *Scara5* ($k_{ME.L}$rank = 15, $k_{ME.L}$rank = 1), *Igfbp5* ($k_{ME.K}$rank = 1), and *Lrig1* ($k_{ME.L}$rank = 10) (**Figure 4G – Figure 4—figure supplement 1F**). Double-labeling experiments using (i) antibody staining against BrdU in wild-type animals that were treated with BrdU followed by a long chase period to generate LRCs and (ii) an antibody against PECAM1 or ISH to detect *Igfbp5* demonstrated that both genes are expressed in LRCs in the incisor mesenchyme and laCL epithelium (**Figure 4J–J"**, and **Figure 4—figure supplement 1G**); the identity of the PECAM1-positive cells in the epithelium is not clear, but

as the epithelium is not vascularized, this marker must label a non-vascular cell type in the epithelium.

We also noted that expression of genes contributing to Modules K-M was predominantly observed in cells of the periodontium (*Figure 4E–G*, and *Figure 4—figure supplement 1E–G'*), which is the tissue responsible for anchoring the tooth to the alveolar bone. While little is known about the periodontal tissue in the incisor, previous studies that mostly focused on molar teeth identified a small number of genes expressed in periodontal cells (*Supplementary file 2*). Comparison with the data obtained from our co-expression analysis showed that most of these factors were associated with Modules K, L, and M, suggesting that this group of modules is driven primarily by gene expression in periodontal cells (*Supplementary file 2*). The expression of these markers in the proximal part of the tooth suggested that these modules may be enriched for candidate stem cell markers and genes required for periodontal tissue maintenance. The relative expression patterns of the modules are presented in *Figure 4O*.

To test whether our analysis could enable functional identification of a stem cell population, we next focused on *Igfbp5*, which was the highest ranked gene contributing to Module K and was previously found to be increased in LRCs in the hair follicle bulge (*Tumbar et al., 2004*). To establish whether *Igfbp5* marks stem cells in the incisor, we generated a tamoxifen-inducible *Igfbp5^iCre-ERT2^* allele that drives expression of *Cre* recombinase without reducing *Igfbp5* expression (*Figure 4—figure supplement 2*), and we bred mice carrying both this allele and the *Ai14* (*R26^RFP^*) reporter allele (*Madisen et al., 2010*) to perform in vivo lineage tracing (*Figure 4K*). 24 hr after tamoxifen treatment of the double-heterozygous animals, RFP expression was detected in cells of the periodontium, the liCL and in the proximal portion and OEE of the laCL, as well as in a number of cells in the alveolar nerve that innervates the incisor. Thus, reporter expression was found in regions of the proximal incisor where we detected *Igfbp5* mRNA expression (*Figure 4J–J"*), and we proceeded to perform lineage analyses using the *Igfbp5^iCre-ERT2^* driver. 14 days after treatment with tamoxifen, the number of RFP-positive cells in the incisor mesenchyme surrounding the cervical loops and within the laCL and distally adjacent epithelium was strongly increased when compared to 24 hr post-induction, suggesting that the originally labeled *Igfbp5*-expressing cells supply new cells in these regions (*Figure 4M*). A similar distribution of RFP-labeled cells was present 1 month after tamoxifen treatment (*Figure 4N*). Interestingly, except for a small number of RFP-labeled cells whose morphology and locations were consistent with a neurobiological identity, RFP-positive cells were absent from the inner dental pulp mesenchyme and remained restricted to the periodontal portion of the incisor mesenchyme. These findings suggest that *Igfbp5*-expressing cells contribute to epithelial and periodontium homeostasis without contributing to maintenance of the pulp compartment.

Notably, *Lrig1*, a marker of stem cells in the intestine and the skin (*Jensen and Watt, 2006*; *Powell et al., 2012*) not previously identified in incisor stem cells, was represented in Module L. Consistent with this observation, *Lrig1* was expressed in the proximal incisor mesenchyme and laCL epithelium but not in pulp cells, odontoblasts or differentiated epithelial cells (*Figure 4G*). ISH yielded a relatively low signal for *Lrig1* when compared to most other genes analyzed for this group of modules, which may suggest transcription at low levels or expression in a subset of cells within the domain. To determine if *Lrig1* is expressed by LRCs, we crossed a tamoxifen-inducible *Lrig1^Cre-ERT2^* mouse line (*Powell et al., 2012*) with the *R26^RFP^* reporter allele. To generate LRCs, *Lrig1^Cre-ERT/+^*;*R26^RFP/+^* newborn mice were injected with BrdU and aged for 8 weeks (*Figure 5A*). Double immunofluorescence assays against BrdU and RFP were performed in specimens from mice chased for 24 hr after tamoxifen administration; the short chase period was used to mark *Lrig1*-positive cells. This experiment confirmed the presence of *Lrig1*-expressing LRCs in the proximal laCL epithelium and the most proximal incisor mesenchyme (*Figure 5B–B"*), indicating that some *Lrig1*-expressing cells are quiescent. Interestingly, in the mesenchyme-derived portion of the incisor, the BrdU-LRC population was comprised of two neighboring subdomains that appear as stripes in the two-dimensional sections: an *Lrig1*-negative inner, or distal, region and an *Lrig1*-positive proximal region (*Figure 5B*). RFP also marked cells in the alveolar nerve, and occasionally single RFP-positive cells were observed within the liCL and in blood vessels. However, RFP was not detected in differentiated cells of the epithelium or within the pulp mesenchyme. Thus, expression of the *Lrig1^Cre-ERT2^* allele was restricted to regions in the incisor where *Lrig1* mRNA is expressed and included some, but not all, BrdU-LRCs (*Figure 4G*).

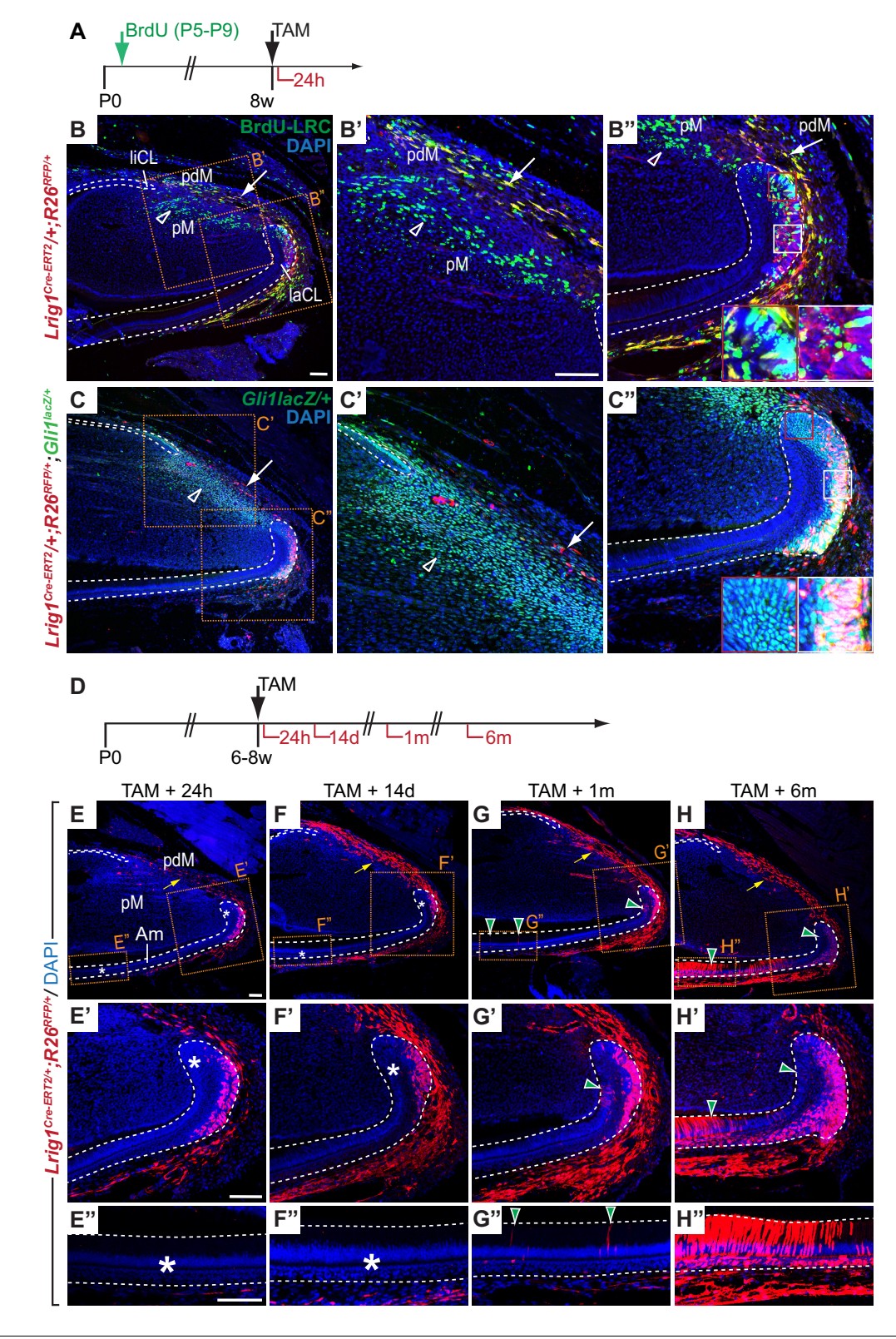

**Figure 5.** *Lrig1* marks stem cell pools in the epithelium and mesenchyme of the mouse incisor. (**A**) Experimental design for testing *Lrig1* expression by label-retaining cells (LRCs). BrdU was administered repeatedly to newborn *Lrig1^Cre-ERT2/+;R26^RFP/+* mice (green arrows), followed by an injection-free period. Cre-mediated recombination of RFP was induced in *Lrig1*-expressing cells 24 hr prior to sacrifice (black arrows). (**B–B"**) A section of the incisor of an adult *Lrig1^Cre-ERT2/+; R26^RFP/+* mouse stained with DAPI (blue), anti-RFP (red) and anti-BrdU (green), 24 hr after administration of Tamoxifen. *Lrig1*

Figure 5 continued

is expressed by LRCs in the lower part of the labial cervical loop (laCL, white box in B" but not by LRCs in the upper laCL (red box in B"). *Lrig1* expression in the proximal mesenchyme divides the mesenchymal BrdU-LRC population into *Lrig1*-negative (green, open arrowhead), pulp mesenchyme (pM)-specific and *Lrig1*-positive (yellow, white arrow), periodontal mesenchyme-specific subdomains. (C–C") *Lrig1*-positive cells are found in incisor regions marked by *Gli1^{lacZ}*-expression. Mesenchymal RFP-positive cells are rare in number when compared to *Gli1^{lacZ}*-positive cells (arrow in C, C') and absent from *Gli1^{lacZ}*-expressing mesenchymal cells distal to the laCL and liCL (open arrowheads). *Lrig1*-positive cells in the laCL express *Gli1^{lacZ}* (white box). RFP expression is absent from a population of *Gli1^{lacZ}*-positive cells in the apical portion of the laCL adjacent to T-A cells (red box). (D) Dosing scheme for in vivo lineage tracing of *Lrig1*-positive cells. (E–H") Lineage tracing of *Lrig1*-expressing cells 24 hr (E, E', E"), 14 days (F, F', F"), 1 month (G, G', G") and 6 months (H, H', H") after Tamoxifen induction. Asterisks indicate the absence of labeled epithelial cells. Green arrowheads denote newly formed labeled progeny in the T-A region or amongst differentiated ameloblasts (Am). Yellow arrows denote increase of labeled cells in the periodontal tissue over time. liCL, lingual cervical loop. Dashed lines delimit epithelium. Scale bars: 100 μm, C as in B; B", C', C" as in B'; F, G, H as in E; F', G', H' as in E'; F", G", H" as in E".

DOI: https://doi.org/10.7554/eLife.24712.013

The following figure supplement is available for figure 5:

**Figure supplement 1.** Periodontal tissue remain quiescent throughout *Lrig1* lineage tracing.

DOI: https://doi.org/10.7554/eLife.24712.014

To definitively determine whether the *Lrig1*-expressing cells in the laCL and the proximal incisor are stem cells that self-renew and give rise to progeny over an extended period, we followed the fate of RFP-labeled cells in tamoxifen-induced *Lrig1^{Cre-ERT2/+};R26^{RFP/+}* mice over chase periods of different lengths (*Figure 5D*). As before, 24 hr after treatment with tamoxifen, we detected RFP-positive cells in the LRC-containing regions of the incisor, including the laCL and the mesenchymal tissue surrounding both CLs (*Figure 5E–E"*); no RFP was detected in the dental pulp mesenchyme between the CLs or in tissues of uninduced *Lrig1^{Cre-ERT2/+};R26^{RFP/+}* control animals (data not shown). 14 days after the initial labeling, *Lrig1*-positive cells remained restricted to the proximal laCL epithelium, and RFP-positive cells were absent from the T-A and differentiated cell regions (*Figure 5F–F"*). Short BrdU chases performed on the lineage tracing specimens revealed that the only proliferative populations were the epithelial and flanking mesenchymal T-A cells, with the periodontium remaining BrdU-negative (*Figure 5—figure supplement 1*).

Next, we wished to determine the relationship between expression of *Lrig1* and *Gli1*, an established marker of both epithelial and mesenchymal stem cells (*Seidel et al., 2010*; *Zhao et al., 2014*). We found that *Lrig1* expression, as reflected by CreER-mediated RFP expression 24 hr after tamoxifen administration, largely overlaps with *Gli1* in *Lrig1^{Cre-ERT2/+};R26^{RFP/+};Gli1^{lacZ/+}* mice; however, *Lrig1* is more restricted in the SR and absent from the most apical portion of the *Gli1^{lacZ}*-positive domain of the laCL (*Figure 5C*). In the mesenchyme, RFP-positive cells were relatively sparse compared to cells expressing *Gli1^{lacZ}*, and these populations do not co-localize in the dental pulp. In the mesenchyme surrounding the CLs, both double-positive or RFP-positive cells that did not express *Gli1^{lacZ}* were detected (*Figure 5C'*). Thus, *Lrig1*-positive cells constitute a subdomain of the *Gli1*-expressing cells in the laCL epithelium, but not in the incisor mesenchyme.

Given the apparent overlap of *Gli1* and *Lrig1* expression, we found it surprising that *Lrig1^{Cre-ERT2/+};R26^{RFP/+}* did not exhibit labeled progeny in differentiated or T-A cells, especially because progeny formation from cells marked by *Gli1*, *Sox2*, or *Bmi1* were detected within days after tamoxifen administration (*Biehs et al., 2013*; *Juuri et al., 2012*; *Seidel et al., 2010*). However, one month post-induction, a small number of labeled cells was present amongst the T-A cells, as well as in the SR, SI and ameloblasts (*Figure 5G–G"*). After 6 months, the number of RFP-positive cells in the T-A region and SR had increased, and the majority of ameloblasts, SI cells and OEE cells were labeled (*Figure 5H–H"*). The extent of RFP labeling in specimens that were analyzed 3 months or 12 months after tamoxifen administration was comparable to the 6 month time point (data not shown). Thus, *Lrig1* is expressed by long-lived stem cells located in the OEE and/or SR of the laCL that give rise to cells of all epithelial lineages over long periods of time. The delay in appearance of labeled progeny in the T-A cell region when compared to lineage tracing performed for *Gli1* and other markers previously tested (*Biehs et al., 2013*; *Juuri et al., 2012*; *Seidel et al., 2010*), together with the absence of *Lrig1*-positive cells from the *Gli1*-expressing population adjacent to the T-A cell region, suggests that *Lrig1* marks a relatively quiescent progenitor pool.

In the proximal incisor mesenchyme, the number of RFP-positive cells strongly increased 14 days after induction with tamoxifen (*Figure 5F*), suggesting that *Lrig1*-positive cells in this region turn over more rapidly when compared to epithelial cells that express *Lrig1*. Similar numbers of RFP-positive cells were observed in the mesenchyme surrounding the CLs in specimens that were analyzed one, 3, 6 or 12 months post-tamoxifen (*Figure 5G* and data not shown). Interestingly, RFP-positive cells were absent from the mesenchyme of the dental pulp, the compartment surrounded by the dental epithelium, but rather were restricted to the periodontal tissue surrounding the epithelial cervical loops on the outer surface of the incisor. Together, these data demonstrate that *Lrig1* marks two distinct pools of stem cells, one in the incisor epithelium that enables renewal of epithelial cell types, such as ameloblasts, and one in the proximal mesenchyme that produces cells contributing to the periodontium. These fate mapping experiments further revealed that the periodontal mesenchyme in the incisor is maintained by a mesenchymal stem cell pool that is separate from the progenitors that maintain the mesenchymal cells of the dental pulp, pointing to the existence of distinct subpopulations of mesenchymal stem cells in the incisor.

## Periodontal stem cell populations marked by Lrig1 and Acta2 partially overlap and mediate renewal of distinct regions of the periodontium

We further interrogated renewal of the periodontium by examining *Acta2*, a gene that had high rankings for Module L (which contained *Lrig1*) based on data from three different microarray probes ($k_{ME.L}$ranks = 120, 150, 200; *Supplementary file 1*). In molars, lineage tracing has shown that *Acta2* is expressed by periodontal progenitors (*Roguljic et al., 2013*). As expected from our co-expression data, *Acta2* expression was found in the periodontal mesenchyme proximal to the epithelial CLs, but not in mesenchymal cells in the pulp (*Figure 4—figure supplement 3B*). Transcription was also detected in perivascular cells in both the periodontal and pulp area, and in a small number of cells in the proximal laCL (*Figure 4—figure supplement 3B'*). In the periodontal compartment, ACTA2 expression included BrdU LRCs (*Figure 4—figure supplement 3C*), but no mRNA or protein was detected in a band of cells directly surrounding the laCL and subtending the adjacent OEE on the labial aspect of the incisor (*Figure 4—figure supplement 3B–C'*). This *Acta2*-negative domain contained a number of LRCs and was *Lrig1*-positive (*Figure 4G – Figure 4—figure supplement 3C'*).

We next performed lineage tracing in *Acta2$^{Cre-ERT2}$;R26$^{RFP}$* mice (*Wendling et al., 2009*) to determine whether *Acta2* is also expressed by periodontal progenitors in the incisor. In contrast to the progenitors in molar teeth, progenitors in the incisor produce periodontal fibroblasts that are constantly renewed and move distally at a similar rate as ameloblasts and odontoblasts (*Smith and Warshawsky, 1976*). Shortly after tamoxifen administration (*Figure 4—figure supplement 3A*), RFP-expressing cells were present in areas where we detected *Acta2* mRNA and protein (*Figure 4—figure supplement 3D*). Similar to what we observed when investigating the fate of *Lrig1*-positive cells, the number of RFP-positive cells in the periodontal mesenchyme increased between the 24 hr and 7 day time points (*Figure 4—figure supplement 3D*). In contrast, the number of labeled cells detected after longer chase periods appeared to remain constant (*Figure 4—figure supplement 3F*) and RFP-positive cells were still present 6 months after tamoxifen treatment (data not shown). These data are consistent with achievement of a steady state and indicate that *Acta2* is expressed by periodontal progenitors in the incisor.

To investigate the relationship between *Lrig1*- and ACTA2-expressing progenitors in the periodontal compartment, we co-labelled cells derived from the *Lrig1*-positive progenitor pool with ACTA2. Interestingly, the progeny of *Lrig1*-positive cells that were labeled three months prior to analysis did not contribute to all regions of the periodontal mesenchyme marked by ACTA2 (*Figure 4—figure supplement 3H*). Whereas expression overlapped in a domain surrounding the ACTA2-negative inner portion of the periodontium, RFP-positive cells were absent from the outermost layer of periodontal tissue, close to the bone. We also assessed co-localization of *Lrig1* descendants with cells expressing N-CAM, a gene that is broadly expressed in the periodontal tissue of the incisor (*Obara and Takeda, 1997*) and that showed strong membership for Module L ($k_{ME.L}$ranks = 39, 206; *Supplementary file 1*). This analysis confirmed that descendants of *Lrig1*-expressing cells contributed to the inner but not outer periodontal mesenchyme (*Figure 4—figure supplement 3I*). In contrast, progeny of cells marked by *Acta2* expression were found in the outer portion of the periodontium but absent from the periodontal tissue directly surrounding the incisor epithelium (*Figure 4—figure supplement 3D–G*). Thus, the inner (near the tooth) region of the periodontal

mesenchyme is preferentially renewed by *Lrig1*-expressing progenitors, whereas the outer (near the bone) region is renewed by *Acta2*-expressing progenitors. Together, these data identify subpopulations of periodontal progenitors, marked by *Lrig1* and *Acta2* expression, that promote renewal of distinct regions of periodontal tissues during incisor homeostasis.

## Discussion

### Gene co-expression analysis elucidates the cellular composition and interplay of biological processes required during renewal of the adult mouse incisor

Here we set out to characterize the cellular composition of the adult mouse incisor and identify cell type-specific markers by analyzing patterns of transcriptional co-variation in a large number of biological replicates. The results from this study provide strong evidence that correlated gene expression patterns are driven by variation in the abundance of distinct cell types and cell states. We identified transcriptional signatures driven primarily by ameloblasts, odontoblasts, Schwann cells, and skeletal muscle cells. The ability to detect a transcriptional signature of a cell type through gene co-expression analysis of intact tissue specimens depends on many factors, including the abundance of the cell type, the distinctiveness of its transcriptome, its anatomical distribution with respect to other cell types, the technology platform, the sampling strategy, and the algorithmic approach (*Oldham, 2014*). Therefore, different cell types will have different signal to noise ratios. For example, ameloblasts and odontoblasts are abundant, differentiated cell types with distinctive transcriptomes that were easily detected with our strategy. In contrast, Schwann cells are much less abundant but co-express a unique set of genes that allowed them to clearly stand out in the incisor co-expression network. Optimization of some of the factors listed above (e.g. RNA-seq analysis of a larger number of samples) should improve the sensitivity of our approach.

We also identified transcriptional signatures related to distinct states of cellular differentiation, including the progenitor state, transit-amplifying state, and cell cycle exit at the onset of differentiation. Among the patterns corresponding to differentiation states, we identified modules representing stem cell progeny far along in their maturation process within distinct compartments as well as compartment-specific modules enriched for genes expressed by progenitors. For example, one newly identified progenitor marker, *Lrig1*, discriminates subpopulations of *Gli1*-positive epithelial progenitors that appear to have different capacities to provide daughter cells for replenishing the organ. By providing an unbiased view of the major transcriptional themes in the adult mouse incisor, our study lays the groundwork for future investigations into molecular interactions that are required to establish or maintain the functional identities of distinct cell types and cell states in this model system. Unbiased gene co-expression analysis of intact biological systems also provides a data-driven framework for studying the effects of perturbations, such as blocking a specific molecular pathway or causing injury. By comparing transcriptional co-variation in perturbed and naive states through differential co-expression analysis, identification of relevant phenotypes that affect specific cell types or cell states can be accelerated.

### Gene co-expression analysis vs. single-cell approaches

Single-cell methods have come of age and hold great promise for gene expression applications. However, technical noise and limited coverage of cells and transcriptomes can constrain the use of this strategy for unbiased characterization of intact biological systems. In contrast, gene co-expression analysis of bulk tissue specimens is a comparatively simple and efficient approach that can reveal the major building blocks of a biological system's transcriptome by analyzing expression patterns that are derived simultaneously from millions of cells. Thus, these two types of analyses can be seen as complementary with regard to resolution and tissue volume analyzed. From a practical perspective, the presence of genes encoding cell-surface proteins in the co-expression modules we have identified will simplify the isolation, purification, and detailed characterization of individual cell populations using single-cell methods, which will be an important next step.

It is also important to note that the microarrays used in this study have limited dynamic range and sample almost exclusively from the protein-coding transcriptome. Therefore, the full picture of gene expression in the adult mouse incisor is likely to be more complex than the initial description

presented here. Future surveys that combine deep sequencing of coding and non-coding transcripts from large numbers of intact tissue specimens and single cells will provide a powerful approach for deconstructing the transcriptional architecture of the incisor and other biological systems. In addition, because our studies here exclusively used two-dimensional section analysis, it will be important in the future to analyze proximal incisor gene expression patterns in three dimensions as well.

## Lineage potential of adult stem cells marked by Igfbp5 or Lrig1

In this study, we discovered several co-expression modules enriched with genes that are predominantly expressed in the LRC-containing regions of the proximal incisor mesenchyme. Amongst the genes with expression profiles most similar to Modules K and L were *Igfbp5* and *Lrig1*. *Lrig1* is expressed in stem cells in the intestine and skin (*Jensen et al., 2009*; *Jensen and Watt, 2006*; *Powell et al., 2012*), and *Igfbp5* is transcribed by LRCs in the hair follicle bulge (*Tumbar et al., 2004*). Interestingly, co-labeling with BrdU-LRCs revealed that expression of both *Igfbp5* and *Lrig1* divides the mesenchymal LRC population into an inner subpopulation that does not express either gene and an outer region where *Igfbp5* and *Lrig1* expression are found. Lineage tracing for *Igfbp5* and *Lrig1* revealed that the outer LRC-containing region of the mesenchyme contains progenitors that contribute specifically to periodontal cell lineages but not mesenchymal cells of the dental pulp. This finding demonstrates for the first time that the inner mesenchyme and the periodontal tissue in this continuously growing tooth are maintained by separate pools of progenitors. The extent to which the *Igfbp5*-positive and *Lrig1*-positive populations overlap will need to be addressed in the future.

Of note, *Lrig1*-expressing cells were shown to contribute specifically to the upper but not the lower portion of the mesenchymally derived dermis of the skin (*Driskell et al., 2013*). This result is intriguing as it parallels our finding that *Lrig1*-expressing mesenchymal cells only contribute to the periodontal tissue but not the dental pulp mesenchyme. Moreover, our analysis of cell proliferation in the *Lrig1* lineage tracing experiments did not identify a specific periodontal T-A cell population, indicating a relatively slow turnover rate of this tissue.

Multiple progenitor pools are known to facilitate coordinated renewal in other organs. For example, in the skin, several stem cell pools exist that contribute, with varying overlap, to the renewal of the interfoliclular epidermis, hair follicles and sebaceous glands (*Jensen et al., 2009*). Although during normal homeostasis the stem cells contribute to their respective compartments, they can be mobilized to regenerate nearby compartments after injury. Whether the *Igfbp5*- or *Lrig1*-expressing periodontal progenitors are able to contribute to the dental pulp lineages following injury remains to be determined. Of note, in a small number of $Lrig1^{Cre-ERT/+};R26^{RFP/+}$ animals chased for one year after tamoxifen treatment, we observed low-level contribution to the odontoblast lineage (data not shown); it is possible that this lineage contribution arose as the result of injury. Alternatively, this increase in plasticity could reflect changes in stem cell number or regeneration capacity that occur in the incisor as animals age.

## Coordination of progeny formation in the mesenchymal and epithelial compartment of the incisor is achieved at the T-A cell level

To achieve proper homeostatic renewal of an organ composed of several tissues, progeny production by stem cell niches in all tissue compartments must be highly coordinated. Indeed, cell-labeling experiments performed in rodent incisors several decades ago showed that ameloblasts, odontoblasts and periodontal fibroblasts move distally at the same rate (*Beertsen and Hoeben, 1987*). The notion that the T-A cell stage serves as an important checkpoint for coordination between tissues is strongly supported by our finding that all factors that contributed to T-A cell specific modules were always expressed in proliferating cells in both the epithelium and mesenchyme and never in only one tissue. Our previous results highlight SHH as a likely signal through which coordination between the different stem cell pools in the incisor is achieved (*Seidel et al., 2010*; *Zhao et al., 2014*). These studies showed that stem cells maintaining the epithelial tissues on the labial and lingual aspect of the tooth, the mesenchymal cell types of the dental pulp including the dentin-forming odontoblasts, and the periodontium are all marked by *Gli1* expression, a hallmark of responsiveness to HH signaling. Similarly, *Bmi1* appears to be expressed by stem cells in all niches of the incisor (*Biehs et al., 2013*), whereas *Sox2* and *Lrig1* mark stem cells only in a restricted set of compartments (*Figure 5*,

(*Juuri et al., 2012*). An interesting open question is whether other members of the SRY-related HMG-box (SOX) family of transcription factors are expressed in the liCL, mesenchymal and periodontal stem cell niches and substitute for the function of SOX2 in these regions, or whether SRY-mediated transcriptional regulation is uniquely required for control of progeny formation from epithelial stem cells housed in the laCL.

Another intriguing discovery is that distinct sets of stem cells, as defined by a constellation of markers uncovered by module analysis, have different properties in the laCL and give rise to unique cell fates in the mesenchyme. *Lrig1*, which encodes a type I transmembrane protein, is specifically expressed by a subset of *Gli1*-expressing cells in the laCL and by stem cells that give rise to periodontal tissues (*Figure 5*). Our lineage tracing analysis further showed that the cells that maintain the lingual incisor epithelium and pulp mesenchyme do not express *Lrig1*. An exciting extension to the finding that *Lrig1* marks a pool of stem cells in the periodontium would be a comparison with the periodontium around molar roots. In both cases, these tissues develop from the dental follicle, wrap the teeth and anchor them to the jaw. Because molars have a finite growth period, in contrast to the continuous growth of the incisor, it would be of interest to determine how the periodontium is maintained in the molar. Additionally, an important future direction will be establishing the molecular function of LRIG1 in the tooth as well as the source and function of the ligand whose signaling it regulates in this system.

In conclusion, the wealth of information gained by our analysis of gene co-expression in the incisor will advance the use of this organ as a model for stem cell-based tissue renewal. More generally, results gained from deconstructing an organ using this transcriptome-focused approach can enable a deeper understanding of the biology of the system. In addition, by comparing gene co-expression networks between different species, important species-specific characteristics of organs can readily be identified. To this end, comparisons of gene co-expression relationships in the developing brains of humans and mice have yielded invaluable insights into transcriptional differences in neural stem cells that have contributed to changes in brain architecture during evolution (*Lui et al., 2014*). Going forward, comparisons of gene co-expression in different organs will help elucidate conserved pathways and general mechanisms that govern tissue renewal from stem cells. Such information can also enhance bioengineering approaches that use stem cells as a starting material for generating tissues for therapeutic purposes.

## Materials and methods

### Animal husbandry

Mice carrying the *Acta2*$^{Cre-ERT2}$ (*Wendling et al., 2009*), *Ai14* (*Madisen et al., 2010*), *Gli1-lacZ* (*Bai et al., 2002*), and *Lrig1*$^{CreERT2}$ (*Powell et al., 2012*) were maintained and genotyped as previously described. 6-week-old female CD1 mice were purchased from Charles River Laboratories and used for generation of tissue samples for microarray analysis. 6–8 week old animals were used for expression analysis and lineage tracing experiments. For generation of label-retaining cells, neonatal mice were injected daily from P5 to P9 with BrdU (5'bromo-2'deoxyuridine) and aged to 8 weeks. For detection of proliferating cells, BrdU was given in a single injection to adult mice 1.5 hr prior to sacrifice. BrdU was administered at 40 μg per gram of bodyweight. Mice were treated with a single dose of 5 mg tamoxifen (in corn oil) given by oral gavage in case of lineage tracing studies, and three doses of 10 mg tamoxifen every other day given in case of ablation experiments. Expression and lineage tracing analyses were performed using specimens from at least three different animals, examined for each functional experiment. All animals were maintained at the UCSF vivarium and the UCSF Institutional Animal Care and Use Committee (IACUC) approved all experiments performed in this study.

### Generation of Igfpb5$^{iCreERT2}$ line

The *Igfbp5*$^{iCreER-T2}$ line was produced at the Jackson Laboratory. To generate the inducible Cre line, a codon optimized Cre recombinase variant (iCre) (*Shimshek et al., 2002*) was fused to a modified ligand-binding domain of human estrogen receptor (ERT2) (*Feil et al., 1997*). To allow for expression from the target locus without disruption of the endogenous allele, an internal ribosome entry sequence (IRES) was placed 5' to the start of the iCreERT2 coding sequence and the cassette was

inserted into the *Igfbp5* sequence after the stop codon. A Frt-flanked neo cassette was placed in the third intron upstream of the final coding exon and the entire construct was flanked by 5' and 3' homology arms. A diphtheria toxin cassette was included for negative selection. The construct was linearized and electroporated into C57BL/6J embryonic stem (ES) cells, subjected to G418 selection and 37 putative targets were identified by loss of native allele (LOA) qPCR screening. Six clones were confirmed by 5' and 3' Southern blot, and three clones with normal karyotypes were injected into albino C57BL/6J blastocysts. Chimeras were bred to C57BL/6J mice and screened for germline transmission of the targeted allele. A single clone was pursued and double confirmed by Southern blot.

## RNA isolation and microarray data generation

Mandibular incisors of 140 wild-type female CD1 mice were isolated as previously described (*Chavez et al., 2014*), and the tissue region proximal to the first occurrence of mineralized dentin on both labial and lingual aspects of the incisor was isolated and stored in RNA-later (Ambion). The tissue level along the proximo-distal axis was readily identified on both lingual and labial aspect of the incisor with a 5.0x magnification based on the color difference between the mineralized tissues. Total RNA was isolated from individual tissue samples using Qiagen's RNeasy kit according to the manufacturer's instructions. To improve the total yield in the final step of the protocol, the eluate was run over the microspin column a second time. Only tissues obtained from left lower incisors were used for RNA extraction. RNA quality and quantity were assessed using a Nanodrop Spectrophotometer and a Bioanalyzer assay, and only the 94 samples with a concentration of >20 ng/µl (average of both assays) and the highest RNA-integrity scores (RIN) of >8.5 were used by the microarray facility. RNA concentrations were confirmed utilizing a ribogreen assay, hybridization to Illumina Mouse Ref 8 v2.0 gene expression BeadChips performed, and initial data analyzed in R with the SampleNetwork function (*Oldham et al., 2012*), which identifies outlying samples, performs data normalization, and adjusts for batch effects. After removing one outlying sample, data were quantile normalized (*Bolstad et al., 2003*) and technical batch effects were assessed. A highly significant batch effect associated with microarray slide was detected and corrected using the ComBat R function (*Johnson et al., 2007*).

## Gene co-expression analysis

Gene co-expression modules were identified in R using a four-step approach as previously described (*Molofsky et al., 2013*; *Lui et al., 2014*). First, pairwise Pearson correlation coefficients (cor) were calculated for all possible pairs of microarray probes (n = 25,697) over all samples (n = 93). Second, probes were clustered using the flashClust implementation of a hierarchical clustering procedure with complete linkage and 1 – cor as a distance measure (*Langfelder and Horvath, 2008*). The resulting dendrogram was cut at a static height of ~0.594, corresponding to the top 1% of pairwise correlations for the entire dataset. Third, all clusters consisting of at least 15 members were identified and summarized by their module eigengene (i.e. the first principal component obtained via singular value decomposition) (*Horvath and Dong, 2008*; *Oldham et al., 2006*). Fourth, highly similar modules were merged if their Pearson correlation coefficients exceeded an arbitrary threshold (0.85). This procedure was performed iteratively such that the pair of modules with the highest correlation >0.85 was merged, followed by recalculation of all module eigengenes, followed by recalculation of all correlations, until no pairs of modules exceeded the threshold. Following these steps, 24 co-expression modules were identified. The strength of module membership ($k_{ME}$) for each probe on the microarray was determined by calculating the Pearson correlation between its expression pattern over all samples with each module eigengene (*Horvath and Dong, 2008*; *Oldham et al., 2008*).

## Module enrichment analysis

Module enrichment analysis with curated gene sets was performed using a one-sided Fisher's exact test in R with gene symbol as a common identifier. Modules were defined as consisting of all unique genes that were positively and maximally correlated with a given module eigengene at a significance threshold of $p<8.11\times10^{-08}$. This threshold corresponds to a Bonferroni-corrected P-value of .05 / (the total number of probes X the total number of modules). Gene Ontology (GO) analysis was

performed using The Database for Annotation, Visualization and Integrated Discovery (DAVID) (*Dennis et al., 2003*). Enriched GO terms for Biological Processes (level 5) were detected and clustered using the Functional Annotation Clustering tool with default parameters.

## Histology, in situ hybridization and immunohistochemistry

Sample preparation and decalcification, hematoxylin and eosin staining, immunofluorescence staining and RNA in situ hybridization were performed as previously described (*Seidel et al., 2010*).

Primers for the ISH probes were designed using Primer-BLAST (http://www.ncbi.nlm.nih.gov/tools/primer-blast), and a BLAST search within the mouse genomic and transcript database was performed for each primer pair in order to ensure specificity. Primer pairs, probe sizes, as well as gene reference sequences are provided in supplementary files. cDNA from mouse incisors was used as a template for PCR amplification, and fragments of interest were cloned into pGEM-T Easy Vector Systems plasmids (Promega). DH5α cells were transformed with 1 µl of plasmid and plated on LB plates with 100 µg/ml ampicillin. Single colonies were picked for overnight amplification in liquid LB, and plasmid DNA was purified using a Plasmid Miniprep kit (Qiagen). 10 µg of plasmid were linearized using 50 u of the appropriate restriction enzyme, and probes were transcribed using SP6, T3, or T7 polymerases together with RNA-DIG labeling mix (Roche). When generating probes to examine the expression of previously identified markers in the incisor system, we noticed that all probes for *Bglap1*, *Bglap2* and *Bglap-rs1* in fact detected transcripts of all three genes, likely as the result of high sequence similarity. Therefore, we designed a probe for in situ hybridization that detects expression of all three genes simultaneously (expression restricted to the odontoblast lineage in the adult incisor). Primer sequences can be found in the *Supplementary file 3*.

Sections used for immunofluorescence were counterstained with DAPI (Vector Laboratories) and mounted in 1% DABCO in glycerol. Signal amplification utilizing the TSA Plus Fluorescein System (Perkin Elmer, NEL741001KT) was performed for detection of BrdU and beta-galactosidase following incubation with appropriate biotinylated secondary antibodies. A Leica-TCS SP5 confocal microscope was used for imaging except for detection of p63 in *Figure 4—figure supplement 1D*. In this case, a Leica DFC500 camera was used with a Leica DM 5000B microscope. For chromatogenic detection of CDKN1A and SOX21, the same sample preparation and antigen retrieval procedures were followed as for immunofluorescence detection. Overnight incubation with the primary antibody was followed by washes in phosphate-buffered saline (PBS), incubation with the appropriate secondary antibody, washes in PBS, incubation with ABC complex (VECTASTAIN Elite ABC HRP Kit, Vector Laboratories, PK6100), washes in PBS, signal detection using a DAB Peroxide substrate kit (Vector Laboratories, SK-4100) according to the manufacturer's instructions, PBS washes and post-fixation in 4% PFA. Slides were mounted with Dako Faramount Aqueous Mounting Media (Dako, S3025). For visualization of BrdU following in situ hybridization, slides were blocked in 5% bovine serum albumin in PBS with 0.1% Tween20 following the color reaction step of the in situ hybridization procedure. Incubation with the primary antibody and subsequent steps were performed as described for detection of CDKN1A and SOX21. Images were acquired using a Leica DFC500 camera on a Leica DM5000B microscope. Information regarding primary and secondary antibodies can be found in *Supplementary file 4*.

## Quantification of proliferating cells

Images of BrdU and DAPI stained sagittal sections of the cervical loop regions of 3 experimental animals and 3 controls were acquired using a Leica-TCS SP5 confocal microscope. BrdU positive cells of the 5 most central sections per specimen were quantified manually using ImageJ, and a Welch two sample t-test was performed.

## Acknowledgements

We thank Sarah Alto and Rebecca d'Urso for assistance with the mouse colony, Steven Garcia for providing mouse muscle RNA for generation of the *Atp2b* in situ probe, and Jeff Bush, Hua Tian, Jason Pomerantz, Amnon Sharir, Jimmy Hu, and members of the Klein and Oldham laboratories for experimental assistance and helpful discussions. This work was funded by NIH R01-DE024988 and R35-DE026602 (ODK), SysCODE interdisciplinary postdoctoral training fellowship RL9-EB008539

and K99-DE024214 (KS), and the UCSF Program for Breakthrough Biomedical Research, which is funded in part by the Sandler Foundation (MCO).

## Additional information

### Funding

| Funder | Grant reference number | Author |
|---|---|---|
| National Institutes of Health | RL9-EB008539 | Kerstin Seidel |
| National Institutes of Health | K99-DE024214 | Kerstin Seidel |
| Sandler Foundation | | Michael C Oldham |
| National Institutes of Health | R01-DE024988 | Ophir D Klein |
| National Institutes of Health | R35-DE026602 | Ophir D Klein |

The funders had no role in study design, data collection and interpretation, or the decision to submit the work for publication.

### Author contributions

Kerstin Seidel, Conceptualization, Formal analysis, Funding acquisition, Validation, Investigation, Visualization, Writing—original draft, Writing—review and editing; Pauline Marangoni, Investigation, Writing—original draft, Writing—review and editing; Cynthia Tang, Bahar Houshmand, Richard L Maas, Investigation; Wen Du, Resources, Writing—review and editing; Steven Murray, Conceptualization, Writing—original draft; Michael C Oldham, Conceptualization, Data curation, Formal analysis, Funding acquisition, Investigation, Writing—original draft, Project administration, Writing—review and editing; Ophir D Klein, Conceptualization, Resources, Formal analysis, Supervision, Funding acquisition, Methodology, Writing—original draft, Project administration, Writing—review and editing

### Author ORCIDs

Pauline Marangoni [iD] http://orcid.org/0000-0002-4355-7322
Ophir D Klein [iD] http://orcid.org/0000-0002-6254-7082

### Ethics

Animal experimentation: This study was performed in strict accordance with the recommendations from the National Institutes of Health Guide for the Care and Use of Laboratory Animals and approved by the Institutional Animal Care and Use Committee at the University of California San Francisco (IACUC protocol AN099613 updated in November 2016 to AN151723).

### Decision letter and Author response

Decision letter https://doi.org/10.7554/eLife.24712.020
Author response https://doi.org/10.7554/eLife.24712.021

## Additional files

### Supplementary files

• Supplementary file 1. Affiliation strengths ($k_{ME}$ values) for all microarray probes with respect to all co-expression modules. The table summarizes probe ID (Probe_ID), HGNC gene symbol (Gene), reference sequence ID (RefSeq_ID), average detection P-value for a probe over all samples (AvgDetPval), number of samples for which the detection P-value for a probe was nominally significant ($p<0.05$ – CountDetPval), indication of whether the probe is a seed gene for the module (Modseed), mean (log2) expression of the probe across all samples (MeanExpr), percentile rank of the probe's mean (log2) expression level among all probes (MeanExprPercentile), module assignment based on expanded module definitions (positive correlation to module eigengene at < Bonferonni-corrected

P-value [8.11e-08]- TopModPosBC_8.11e-08), module assignment based on expanded module definitions (positive correlation to module eigengene at <FDR-corrected P-value [0.012] – TopMod-PostFDR_0.012), and for each of the modules, correlation of the probe's expression pattern across all incisor samples the eigengene (A-X.kME), and corresponding P-value for $k_{ME}$ (A-X-pval). That information is provided for the 25,697 probes of the Unique Illumina HT12-v4 microarray used.
DOI: https://doi.org/10.7554/eLife.24712.015

• Supplementary file 2. Module association of factors expressed in the periodontal tissue compartment. Microarray Probe ID information is listed for genes with previously demonstrated periodontal tissue-specific expression. For each detected transcript co-expression module association as derived from the $k_{ME}$ table and a ranking score for specificity within the associated module are given. In case of representation by multiple microarray probes and assignment to more than one module ranking scores are listed for all modules. Factors associated with module cluster K-L-M are highlighted (orange). (a) Expression also in dental pulp. (b) Expression also in odontoblasts. (c) Probe on array detects *Col4a2*. (d) Probe on array detects *Inta11*. (e) Transcript detected only in 9/94 samples (average deection p-value: 0.304738). [f] Expression also in ameloblasts.
DOI: https://doi.org/10.7554/eLife.24712.016

• Supplementary file 3. Table of DIG-labeled probe information. For each gene, the forward and reverse primer sequences are available, along with probe size and Reference Sequence.
DOI: https://doi.org/10.7554/eLife.24712.017

• Supplementary file 4. List of the primary and secondary antibodies used. This table provides the list of primary and secondary antibodies used for immunofluorescent staining analysis, with their catalog number and working dilution.
DOI: https://doi.org/10.7554/eLife.24712.018

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
