## [Decision Letter]

Thank you for submitting your article "Resolving stem and progenitor cells in the adult mouse incisor through gene co-expression analysis" for consideration by *eLife*. Your article has been reviewed by two peer reviewers, and the evaluation has been overseen by a Reviewing Editor and Marianne Bronner as the Senior Editor. The following individual involved in review of your submission has agreed to reveal his identity: Igor Adameyko (Reviewer #3).

The reviewers have discussed the reviews with one another and the Reviewing Editor has drafted this decision to help you prepare a revised submission.

Summary:

The reviewers agreed that your paper is a very clear analysis of a stem cell population in the continuously growing incisor and that it provides an impressive dataset to classify different cells within a tooth, which could be utilised by a number of researchers. The reviewers also noted that the authors make a good attempt to highlight problems with their data and the methods used and to compare their work to the use of other strategies, which is refreshing.

Essential revisions:

1) There were several suggestions regarding the microarray data:

A) The microarray that was used does not include all key genes. The authors highlight problems in sensitivity given that genes which should have been identified were not. The full picture is therefore likely to be more complex than shown. The authors should indicate this in the manuscript.

B) The hierarchical clustering shown in Figure 3A and Figure 4A is very interesting. Still, it would nice to know how the individual clusters (with assigned letters) correspond to the cell types: fully or not fully (at least this might be discussed). Of course, it is always safer to talk about the tissue regions with highly co-expressed genes. Still, such co-expression means that these genes are likely expressed in the same or in very similar cells. Some technical discussion on the noise/signal ratios and cell type related interpretation of clusters would be beneficial.

2) In Figure 5 authors show and claim that *Lrig1* marks polls of epithelial and also mesenchymal stem cells in the incisor. However, pulp cells and odontoblasts do not form from *Lrig1^+^* population of mesenchymal cells. It is better to write this part and the figure legend title more clearly to point out that *Lrig1^+^* cells produce/self-renew only dental follicle/periodontal tissues.

3) It would be good to show the in-depth analysis of proliferation in traced dental follicle/periodontal populations. Do these cells divide only at the CL regions? Or they divide only at the opening in the center between CLs? Is there a TA population resulting from *Lrig1^+^* cells in case they are classical stem cells, and is there a niche? Do traced cells undergo apoptosis closer to the tooth apex similar to ameloblasts? Basically, it will good to answer if *Lrig1^+^* cells satisfy the classical concept of stem cells positioned in a niche environment.

---

## [Author Response]

*Essential revisions:*

*1) There were several suggestions regarding the microarray data:*

*A) The microarray that was used does not include all key genes. The authors highlight problems in sensitivity given that genes which should have been identified were not. The full picture is therefore likely to be more complex than shown. The authors should indicate this in the manuscript.*

We appreciate this suggestion and have provided more discussion about this issue in the revised manuscript. We discuss more thoroughly the limitations imposed by the dynamic range of the microarray platform that we used and propose next steps such as broader transcriptomic studies of coding and non-coding transcripts from intact tissue along with single-cell approaches. We agree that the full extent of gene expression in the adult mouse incisor is likely to be more complex than our current understanding, but anticipate that our study will provide a strong foundation to build upon moving forward.

*B) The hierarchical clustering shown in Figure 3A and Figure 4A is very interesting. Still, it would nice to know how the individual clusters (with assigned letters) correspond to the cell types: fully or not fully (at least this might be discussed). Of course, it is always safer to talk about the tissue regions with highly co-expressed genes. Still, such co-expression means that these genes are likely expressed in the same or in very similar cells. Some technical discussion on the noise/signal ratios and cell type related interpretation of clusters would be beneficial.*

We agree with this critique and have revised the manuscript to clarify which clusters correspond to which cell types. In addition, we have added technical discussion about the signal/noise ratio and cell type related interpretation of clusters, as suggested.

*2) In Figure 5 authors show and claim that Lrig1 marks polls of epithelial and also mesenchymal stem cells in the incisor. However, pulp cells and odontoblasts do not form from Lrig1+ population of mesenchymal cells. It is better to write this part and the figure legend title more clearly to point out that Lrig1+ cells produce/self-renew only dental follicle/periodontal tissues.*

We agree with this suggestion and have modified the relevant part of the text as proposed by the reviewers, explicitly stating that *Lrig1^+^* cells contribute to follicle and periodontium but not pulp.

*3) It would be good to show the in-depth analysis of proliferation in traced dental follicle/periodontal populations. Do these cells divide only at the CL regions? Or they divide only at the opening in the center between CLs? Is there a TA population resulting from Lrig1+ cells in case they are classical stem cells, and is there a niche? Do traced cells undergo apoptosis closer to the tooth apex similar to ameloblasts? Basically, it will good to answer if Lrig1+ cells satisfy the classical concept of stem cells positioned in a niche environment.*

To address these questions, we have performed BrdU injections 1.5 h before sacrifice of animals in which *Lrig1* lineage tracing had been induced. It appears that the only proliferative cell populations found in the incisor in those conditions are the epithelial and mesenchymal T-A cell populations that have already been described. The periodontium remains BrdU negative, which indicates that this population is quiescent, consistent with the slow turnover of this tissue. These data have been added to Figure 4—figure supplement 1, and the Results (“Modules K-M are enriched for candidate stem cell markers and markers specific to periodontal tissues”) and Discussion (“Lineage potential of adult stem cells marked by *Igfbp5* or *Lrig1*”) have been updated to reflect our answer to the reviewers’ questions.